# Design, Synthesis, and Antisickling Investigation of a Thiazolidine Prodrug of TD-7 That Prolongs the Duration of Action of Antisickling Aromatic Aldehyde

**DOI:** 10.3390/pharmaceutics15112547

**Published:** 2023-10-28

**Authors:** Rana T. Alhashimi, Tarek A. Ahmed, Lamya Alghanem, Piyusha P. Pagare, Boshi Huang, Mohini S. Ghatge, Abdelsattar M. Omar, Osheiza Abdulmalik, Yan Zhang, Martin K. Safo

**Affiliations:** 1Department of Pharmaceutical Chemistry, Faculty of Pharmacy, King Abdulaziz University, Alsulaymanyah, Jeddah 21589, Saudi Arabia; ralhashimi@kau.edu.sa (R.T.A.); asmansour@kau.edu.sa (A.M.O.); 2Department of Pharmaceutics, Faculty of Pharmacy, King Abdulaziz University, Alsulaymanyah, Jeddah 21589, Saudi Arabia; 3Centre for Artificial Intelligence in Precision Medicines, King Abdulaziz University, Alsulaymanyah, Jeddah 21589, Saudi Arabia; 4Department of Medicinal Chemistry and The Institute for Structural Biology, Drug Discovery and Development, School of Pharmacy, Virginia Commonwealth University, Richmond, VA 23298, USA; alghaniml@vcu.edu (L.A.); pagarepp@vcu.edu (P.P.P.); bhuang2@vcu.edu (B.H.); msghatge@vcu.edu (M.S.G.); yzhang2@vcu.edu (Y.Z.); msafo@vcu.edu (M.K.S.); 5Division of Hematology, The Children’s Hospital of Philadelphia, Philadelphia, PA 19104, USA; abdulmalik@email.chop.edu

**Keywords:** thiazolidine prodrug, aromatic aldehyde, oxygen affinity, sickling inhibition, hemoglobin modification, sickle cell disease

## Abstract

The synthetic allosteric effector of hemoglobin, TD-7 has been investigated as a potential therapeutic agent for the treatment of sickle cell disease. The pharmacologic activity of TD-7 is due to formation of a Schiff-base interaction between its aldehyde group and the two N-terminal αVal1 amines of hemoglobin, effectively inhibiting sickling of red blood cells. However, TD-7 faces a challenge in terms of poor oral bioavailability due to rapid in-vivo oxidative metabolism of its aldehyde functional group. To address this shortcoming, researches have explored the use of a L-cysteine ethyl ester group to cap the aldehyde group to form a thiazolidine aromatic aldehyde prodrug complex, resulting in the improvement of the metabolic stability of this class of compounds. This report details the synthesis of a thiazolidine prodrug of TD-7, referred to as Pro-7, along with a comprehensive investigation of Pro-7 functional and biological properties. In an in-vitro Hb modification and Hb oxygen affinity studies using normal whole blood, as well as erythrocyte sickling inhibition using sickle whole blood, Pro-7 exhibited a gradual onset but progressive increase in all activities. Additionally, in-vivo pharmacokinetic studies conducted with Sprague Dawley rats demonstrated that Pro-7 can undergo hydrolysis to release TD-7. However, the blood concentration of TD-7 did not reach the desired therapeutic level. These findings suggest that the incorporation of the L-cysteine ethyl ester group to TD-7 represents a promising strategy to enhance the metabolic stability of aromatic aldehydes that could lead to the development of a more effective drug for the treatment of sickle cell disease.

## 1. Introduction

Sickle cell disease (SCD) is caused by a single point mutation where Glutamic acid (βGlu6) at the 6th position of the β-chain of normal hemoglobin (HbA) is changed to valine (βVal6), forming abnormal hemoglobin (HbS) [1,2]. At low oxygen (O_2_) tensions, this mutation causes HbS to polymerize into rigid insoluble fibers, primarily by hydrophobic interaction between HbS molecules, resulting in sickling of erythrocytes or red blood cells (RBCs) [3,4,5,6,7,8,9,10]. The rigid and brittle sickled RBC leads to RBC hemolysis, RBC endothelial adhesion, deficiency of nitric oxide, inflammation, painful vaso-occlusive crises (VOC), multi organ damage, and eventually premature death [3,5,6,7,9,11,12,13,14]. The current medications for SCD are hydroxyurea [15,16], Endari [17,18], Crizanlizumab [19], and Voxelotor [20,21,22,23].

Voxelotor is the first in class aromatic aldehyde molecule that was approved in 2019 by the FDA for the treatment of SCD [20,21,22,23]. The antisickling activity of aromatic aldehydes is primarily due to the compounds ability to form Schiff-base interaction with the two N-terminal αVal1 amines of Hb at the α-cleft of the protein, which in addition to other protein contacts, lead to stabilization and increase concentration of the non-polymer forming R-state oxygenated HbS; subsequently preventing hypoxia-induced HbS polymerization and RBC sickling [12,20,24]. Other aromatic aldehydes, including vanillin, 5-hydroxymethylfurfural (5-HMF) and TD-7 (Figure 1) have also been studied for their antisickling potentials [12,20,24,25,26,27]. 5-HMF underwent phase clinical I/II studies [12,28,29], while vanillin and TD-7 were studied in the preclinic for the treatment of SCD [12,30]. TD-7 is a second-generation aromatic aldehyde, obtained by incorporating an alcohol substituted methoxypyridine to the benzaldehyde ring of a vanillin analog (Figure 1), resulting in TD-7 exhibiting several fold sickling inhibition potency over vanillin [27]. In contrast to Voxelotor, which demonstrated satisfactory oral bioavailability, vanillin, 5-HMF and TD-7 suffered from low oral bioavailability [12,31,32,33], ultimately leading to termination of their respective studies. Voxelotor metabolic stability is due to an *ortho* hydroxyl group (relative to the aldehyde moiety) that provided protection against unwanted metabolic breakdown [12]. Vanillin, 5-HMF and TD-7 lack this hydroxyl protection. In an effort to solve the problem of aldehyde metabolism, Zhang, et al. prepared a prodrug of vanillin (MX-1520) (Figure 1), where the aldehyde functional group was protected by L-cysteine ethyl ester group (thiazolidine group), which gradually decomposes over a period of time to release the parent active vanillin [34]. MX-1520 resulted in a much better oral bioavailability compared to vanillin. MX-1520 has provided insights into how to decrease the metabolism and increase the duration of action of aromatic aldehyde pharmacophores.

In this study, we have used the thiazolidine prodrug approach, where the aldehyde group was protected via a coupling reaction with L-cysteine ethyl ester to form a thiazolidine complex, enhancing the metabolic stability of TD-7. The synthesized thiazolidine TD-7 prodrug, referred to as Pro-7 (Figure 1), is anticipated to hydrolyze in-vivo, releasing the active parent TD-7, which can subsequently engage in Schiff-base interactions with Hb to increase HbS affinity for oxygen and prevent sickling of red blood cells.

## 2. Materials and Methods

### 2.1. Materials

Eudragit L100 was obtained from Evonik Industries AG-Werk Rohm (Darmstadt, Germany). Dichloromethane was purchased from Prolabo (Paris, France). All other reagents used in the syntheses and functional/biological assays were purchased from Sigma-Aldrich corporation (St. Louis, MO, USA) and Fisher Scientific (Pittsburg, PA, USA) and used without purification. TD-7 was synthesized as previously published [27].

### 2.2. Study Approvals

At Virginia Commonwealth University (VCU), normal whole blood (AA) was collected from adult donors (>18 years) after informed consent, in accordance with regulations of the IRB for Protection of Human Subjects (IRB #HM1) by the Institutional Review Board at VCU. At the Children’s Hospital of Philadelphia (CHOP), leftover blood samples from individuals with homozygous sickle cell (SS) who had not been recently transfused, were obtained and utilized based on an approved IRB protocol (IRB# 11-008151) by the Institutional Review Board at CHOP, with informed consent. All experimental protocols and methods were performed in accordance with institutional (VCU and CHOP) regulations.

### 2.3. Chemistry Synthesis of Pro-7, the Prodrug of TD-7

^1^H-NMR and ^13^C-NMR spectra were obtained on a Brucker 400 MHz spectrometer using tetramethylsilane (TMS) as an internal standard. Peak positions are given in parts per million (δ). Column chromatography was performed on silica gel (grade 60 mesh; Bodman Industries, Aston, PA, USA). Routine thin-layer chromatography (TLC) was performed on silica gel GHIF plates (250 μm, 2.5 × 10 cm; Analtech Inc., Newark, DE, USA). Infrared spectra were obtained on a Thermo Nicolet iS10 FT-IR.

Synthesis of 1-(2-(3-((6-(hydroxymethyl)pyridin-2-yl)methoxy)-5 methoxyphenyl)thiazolidin-4-yl)propan-1-one, (Pro-7).

TD-7 prodrug, Pro-7 was synthesized following the published procedure for synthesizing the thiazolidine prodrug of vanillin, MX-1520 [34]. To a suspension of 2-((6-(hydroxymethyl)pyridin-2-yl)methoxy)-5-methoxybenzaldehyde (TD-7) (0.706 mmol) in ethanol was added a solution of L-cysteine (1.76 mmol) dissolved in ethanol containing ethylenediamine, EDA (1.76 mmol). The reaction mixture was stirred at room temperature for one hour until no starting materials were left. The reaction mixture was quenched with water, and the resulting white precipitate was subsequently isolated through filtration, followed by several washing cycles with water. The dried material was crystallized using methanol, and the crystals filtered using vacuum drying with a final product yield of 40.9%. IR (Diamond, cm^−1^): 3324.30, 3211.91, 2972.14, 2906.45, 1734.69, 1601.29, 1577.85, 1495.98, 14.55.73, 1372.02, 1313.64, 1271.63, 1212.81, 1073.99, 1036.22; ^1^H NMR (400 MHz, CDCl_3_): δ 1.28–1.34 (m, 3H, C26-H), [2.99–3.07 (m, 1.1H), 3.17 (dd, *J*_1_ = 8.0 Hz, *J*_2_ = 4.0 Hz, 0.53H), 3.43 (dd, *J*_1_ = 8.0Hz, *J*_2_ = 8.0 Hz, 0.57H,) C11-H,C13-H], 3.7 (d, *J* = 4, 3H, C28-H), ), 4.25 (m, 2H, C25-H), 4.76 (d, *J* = 4 Hz, 2H, C9-H), 5.25 (d, 2H, C20-H) [3.95 (dd, *J*_1_ = 8.0 Hz, *J*_2_ = 8.0 Hz, 0.56H) C11-H, C13-H )], [5.88 (s, 0.5H), 6.1 (s, 0.5H) C7-H], [6.7–6.8 (m, 2H), 7.08 (t, *J* = 3.2, 1H) 7.314–7.17 (m, 1H) 7.48 (d, *J* = 8 Hz, 0.5H) 7.56 (d, *J* = 8, 0.5H) 7.67–7.74 (m, 1H) C3-H, C4-H, C6-H, C15-H, C16-H and C19-H]; ^13^C NMR (400 MHz, CDCl_3_): δ 14.1 CH_3_, 37.7, 39.01, 55.7, 61.57, 63.98, 65.07 65.8, 67.86, 70.99, 71.47, 99.84, 100.2, 104.76, 105.34 112.39, 112.78, 113.85, 114.09, 119.27, 119.99, 127.67, 155.7, 132.23, 137.55, 149.33, 150.05, 153.94, 154.1, 156.2, 158.32, 171.5; Mass spectrum *m*/*z.* found 427.1295 [M+Na]^+^; calcd, (427.1406) [M+Na]^+^. C_20_H_24_N_2_O_5_S (404.4810). M.P. 100.8–101.5 °C.

### 2.4. In-Vitro Stability Studies

The stability of Pro-7 and the control TD-7 in different pH buffers (PBS buffer at pH 7.4, KCl·HCl buffer at pH 2.0, and Tris buffer at pH 8.0) was evaluated over time as previously reported [35]. The ionic strength of all buffers was kept constant at 150 mM. A 200 mM stock solution of each compound was prepared using anhydrous DMSO. 1 mM solutions of each of the test compounds TD-7 and Pro-7 were prepared in duplicate for each of the three different pH buffers. The solutions in Eppendorf tubes were subsequently incubated at 37 °C for 24 h with continuous shaking at 140 rpm. At defined time points (1, 2, 3, 6, 24, 28, 48, 72, and 96 h) a 1 mL aliquot sample was taken to measure the absorbance of the aldehyde peak at the wavelength 349 nm using UV-visible spectrophotometer (Hewlett-Packard, Agilent Technologies (New Delhi, India).

### 2.5. In-Vitro Time-Dependent Hb Modification Studies Using Human Normal Whole Blood

Pro-7 and the control TD-7 were used to conduct time-dependent Hb modification (Hb adduct formation) studies using normal whole blood as previously reported [11,27]. A 200 mM stock solution of each compound was made with DMSO. In a 96-well 1.2 mL polypropylene deep-well plate (Thermo Scientific, Hampshire, UK), 2.0 mM concentration of each compound was added to 600 μL of whole blood (30% hematocrit), which was then incubated at 37 °C for 24 h with shaking (at 140 rpm). A control sample was supplemented only with an equivalent volume of DMSO. At different time points (0–24 h), 50 μL aliquots of this mixture were withdrawn from each well using a multichannel pipette and added to respective tubes containing 50 μL of sodium cyanoborohydride (NaBH_3_CN) and sodium borohydride (NaBH_4_) mixture (1:1 *v*/*v* 50 mM stock) to terminate the Schiff-base reaction, fix the Schiff-base Hb adducts and reduce the free reactive aldehyde. After mixing, the tubes were stored immediately at −80 °C until ready for analysis using cation-exchange HPLC (Hitachi D-7000 Series, Hitachi Instruments, Inc. San Jose, CA, USA) for Hb-compound adduct formation. The observed Hb adducts in %Hb modification were plotted as a function of time (h).

### 2.6. In-Vitro Time-Dependent Oxygen Equilibrium Studies Using Human Normal Whole Blood

An oxygen equilibrium curve study was conducted in a time-dependent manner to determine the effect and duration of Pro-7 on Hb affinity for oxygen (P_50_ shift) following standard procedure [36]. Pro-7, along with the controls TD-7 and DMSO were used for the studies with normal whole blood sample (30% hematocrit). Each compound was incubated with the blood at a final concentration of 2 mM, and aliquot samples were taken at 1, 3, 6, 12 and 24 h and further incubated in IL 237 tonometer (Instrumentation Laboratories, Inc., Lexington, MA, USA) for 10 min at 37 °C, and allowed to equilibrate at oxygen tensions 6, 20, and 40 mmHg. The samples were then aspirated into a Radiometer ABL 800 Automated Blood Gas Analyzer (Copenhagen, Denmark ) to determine the pH, partial pressure of CO_2_ (pCO_2_), partial pressure of oxygen (pO_2_), and Hb oxygen saturation values (SO_2_). The measured values of pO_2_ (mmHg) and SO_2_ at each pO_2_ value were then subjected to a non-linear regression analysis using the program Scientist (Micromath, Salt Lake City, UT, USA) to estimate P_50_ as a measure of Hb oxygen affinity as previously reported [36].

### 2.7. In-Vitro Time-Dependent Sickling Inhibition Studies Using Homozygous Sickle Cell (SS) Blood

Red blood cell sickling inhibition studies with Pro-7 and the control TD-7 was conducted using previously reported procedure [11]. Blood suspensions from subjects with homozygous SCD (hematocrit 20%) were incubated under air in the absence or presence of test compounds at 37 °C for 1 h to ensure that binding has attained equilibrium. Following, the suspensions were incubated under hypoxic conditions (2.5% O_2_/97.5% N_2_) at 37 °C for 6 h. Aliquot samples were fixed with 2% glutaraldehyde solution without exposure to air, and then subjected to microscopic morphological analysis.

### 2.8. Development of Enteric-Coated Beads and Intravenous (IV) Formulations

#### 2.8.1. Solubility Study

The solubility of Pro-7 was tested in water, ethanol, methanol, dichloromethane, chloroform, and acetone. An excess amount of the Pro-7 was added to a known volume of each solvent in a screw cap vial. Each vial was kept in a shaking water bath at 25 °C for 48 h. The content of each vial was filtered using an Acrodisc^®^ syringe filter of 0.45 mm (Port Washington, NY, USA), and the concentration of Pro-7 determined spectrophotometrically at 296 nm.

#### 2.8.2. Preparation of Enteric-Coated Microspheres

In this study, enteric-coated microspheres were prepared using the previously published emulsion solvent evaporation technique with slight modifications [37,38]. A compound-to-polymer “Eudragit L100” ratio of 1:5 was used. The known weight of the Pro-7 and Eudragit L100 were dissolved in an acetone ethanol mixture (2:1), and the polymeric compound mixture was added dropwise to 100 mL of paraffin oil containing 3 g of span 80, and the mixture stirred at room temperature until complete evaporation of the solvent. The mixture was set aside overnight for complete curing to allow the spheres to precipitate. The paraffin oil was decanted and n-hexane was added to the precipitated microspheres. The mixture was filtered, washed several times with n-hexane, and the filtered spheres left to dry at room temperature. The dried enteric-coated beads were weighed and compared to the original weight of Pro-7 and polymer.

To identify the Pro-7 content in the prepared microspheres, known weight “30 mg” of the microspheres was dissolved in 50 mL ethanol, and the mixture subjected to continuous stirring over a magnetic stirrer for 30 min. Following, the mixture was filtered and the concentration of Pro-7 was determined spectrophotometrically.

#### 2.8.3. Stability of the Prepared Microspheres in Buffer pH 1.2 and 7.4

The stability of the prepared enteric-coated microspheres was tested in three different buffers namely; pH 1.2, 5.5 and 7.4. Briefly, 50 mg of the prepared microspheres were added to 100 mL of each buffer and left shaking on a magnetic stirrer for 2 h at 100 rpm. The mixture was taken, filtered and the Pro-7 content was determined spectrophotometrically against blank of the same buffer.

#### 2.8.4. Preparation of IV Formulation

Self-nanoemulsifying drug delivery system (SNEDDS) was used to prepare Pro-7 loaded intravenous (IV) formulation. Briefly, oleic acid as an oil, tween 80 as a surfactant, and polyethylene glycol (PEG) 400 as a cosurfactant were used in a concentration of 15%, 10%, and 75%, respectively. One gram of SNEDDS was prepared by accurately weighing the calculated amount of each component in an Eppendorf tube. The mixture was vortexed for 30 s to a homogenous dispersion. The total weight of the oil, surfactant and cosurfactant in the prepared SNEDDS mixture always added to 100%.

The globule size and polydispersity index (PDI) of the prepared SNEDDS were determined after dilution with distilled water (1:10) and vortex mixing using Malvern Zetasizer Nano ZSP, Malvern Panalytical Ltd. (Malvern, UK). Dynamic light scattering with non-invasive backscatter optics was used to measure the size. An average of three readings was recorded.

To determine the drug solubility in the prepared SNEDDS formulation, excess amount (approximately 300 mg) of Pro-7 was added to 4 mL of the prepared SNEDDS. The mixture was vortex and kept in a shaking water bath at 25 °C for 48 h, and then filtered using Acrodisc^®^ syringe filter (0.45 mm) and the concentration of Pro-7 determined spectrophotometrically at 296 nm.

### 2.9. In-Vivo Pharmacokinetic Studies in Rat

Investigations were carried out to examine the pharmacokinetic profiles of Pro-7, including the enteric-coated microspheres and the intravenous (IV) formulation using a one-period parallel design. The study design and conduct adhered to ethical guidelines and regulatory standards to ensure the welfare of the animals. The study was also conducted in strict adherence to internationally recognized guidelines for ethical research and regulatory standards. The principles of Good Clinical Practice (GCP), the guidelines set forth by the International Conference on Harmonization (ICH), and the rigorous standards of the European Medicines Agency (EMA) served as the framework for the study’s execution. The protocol for the animal study underwent a thorough evaluation and received formal approval from the Research Ethics Committee at the Faculty of Pharmacy, King Abdulaziz University, Saudi Arabia.

#### 2.9.1. Animal Housing

Male Sprague Dawley rats, each with an average weight of 250 g were housed in a controlled facility with a 12-h light/dark cycle, at constant temperature of 25 °C, with continuous access to both food and water throughout the study. All animals were acclimated to standard housing for at least 7 days. The animals were separated into distinct groups (*n* = 5). Group I was treated with a single PO (50 mg/kg body weight) dose, while Group II was administered a single IV (25 mg/kg body weight) dose of Pro-7. Additional three control groups were included to validate and enhance the robustness of our findings. These include a normal control group without treatment, a control IV administration group that was administered non-medicated SNEDDS formulation, and a control oral administration group that was given non-medicated oral formulation.

The oral formulation was suspended in 1% aqueous carboxymethyl cellulose. Blood samples from the jugular vein, at 0.25 (for IV only), 0.5, 2, 4, 8, 10, 12, and 24 h were collected into EDTA tubes. To determine the concentration of TD-7 in the whole blood, aliquots (45 µL) of blood samples were mixed with 20 µL of warfarin (used as IS) in a concentration of 1 µg/mL and 500 µL of 1% formic acid in acetonitrile. The vials were centrifuged for 5 min at 10,000 rpm, and 2 µL of the resulting solution was analyzed by LC-MS/MS.

#### 2.9.2. Chromatographic Conditions

The concentration of both TD-7 and Pro-7 in the collected blood samples were quantified using LC–MS/MS method utilizing LLS/DMPK/005 Shimadzu-8045 LC-MS/MS instrument equipped with Lab solution 6.8 software. Linearity of the TD-7 and Pro-7 assay methods was verified within the concentration range 8.21–3776 and 20.4–9396 ng/mL, respectively. A regression coefficient (R^2^) of more than 0.993 was obtained. The analytical method was validated in accordance with the FDA Bio-analytical Method Validation Guidelines 2003. Chromatographic separation was conducted on Atlantis C18 (4.6 × 50 mm, 3.5 µm) at 30 °C. The injection volume was 2 µL, the flow rate was adjusted at 0.7 mL/min, the total run time was 2.5 min, and the mobile phase consisted of 0.1% formic acid in acetonitrile:10 mM aqueous ammonium formate (70:30 *v*/*v*). The retention times for TD-7, Pro-7 and IS were 1.047, 1.178, and 1.503 min, respectively. The limit of quantification for TD-7 and Pro-7 was established at of 8.21 and 20.4, respectively.

#### 2.9.3. Pharmacokinetic Analysis

Pharmacokinetic parameters were calculated using non-compartmental analysis with Kinetica TM (version 4, Thermo Electron Corporation, Waltham, MA, USA) software. The following parameters were determined: Maximum plasma concentration (C_max_), Elimination half-life (t½), Area under the plasma concentration-time curve from time 0 to the last measurable concentration (AUC_0–t_), Area under the plasma concentration-time curve from time 0 to infinity (AUC_0–∞_), Clearance (Cl), and Volume of distribution (V_d_).

The t½ was determined as 0.693 divided by the elimination rate constant. The apparent total body clearance was calculated by dividing the administrated dose (adjusted for the fraction of drug absorbed in case of oral administration) by the dose itself. The apparent volume of distribution was estimated by multiplying the total body clearance by the mean residence time of the drug. The fraction of the oral dose absorbed (F%) was calculated as (AUC oral/AUC IV × 100) [39].

#### 2.9.4. Statistical Analysis

All data were expressed as mean ± SD and subjected to statistical analysis using GraphPad Prism software, version 8 (GraphPad Inc., La Jolla, CA, USA). A *p*-value less than 0.05 was considered significant.

## 3. Results and Discussion

### 3.1. Protection of TD-7 by L-Cysteine Ethyl Ester 

Based on the findings of the thiazolidine prodrug of vanillin [34], we decided to employ a similar strategy of protecting the active aldehyde group of TD-7 from undue metabolism by coupling the aldehyde group with L-cysteine ethyl ester group to from thiazolidine prodrug of TD-7. This approach is expected to improve TD-7 oral bioavailability. The thiazolidine TD-7 complex, Pro-7 was synthesized in two steps as shown in Figure 2. The first step is the synthesis of TD-7 as previously reported [27]. The second step involved the incorporation of L-cysteine ethyl to TD-7 forming a thiazolidine ring through a nucleophilic addition reaction between L-cysteine (an aminothiol) and the aromatic aldehyde of TD-7. This ring-closure process introduced a novel chiral center in an unregulated fashion, resulting in the formation of Pro-7 as a mixture of diastereomer, with a yield of 40.9%. The reaction was carried out under basic conditions at room temperature using anhydrous ethanol as the solvent. Pro-7 was characterized by ^1^HNMR, ^13^CNMR, HRMS and IR. The ^1^HNMR spectra displayed the anticipated characteristic signals associated with the generated thiazolidine ring. Specifically, in addition to the singlet peaks appearing around 5.8 ppm and 6.2 ppm, corresponding to the C7-H signals of each diastereomer, the spectra exhibited a series of pairs of doublet of doublets (dd) at a range from 2.9 ppm to 4.0 ppm for the C11-H and C13-H signals (Appendix A). Our findings are highly similar to a published compound with thiazolidine ethyl ester [40]. The use of HPLC to determine the purity of Pro-7 did not work due to its instability as the compound degraded on the C-8 column. However, the purity of the compound was verified by other methods (Appendix A). Multiple TLC runs with different mobile phases (hexane, ethylacetate and methanol) showed a clean single spot corresponding to the desired product. HRMS analysis identified a single mass in the injected sample, directly linked to the desired substance (Appendix A). The narrow melting point range of 100.8–101.5 °C is also consistent with the substance’s purity. Most significantly, the ^1^HNMR and ^13^CNMR spectroscopy confirmed the presence of all proton and carbon peaks respectively with correct integration for the substance, with no detectable impurities. It is worth noting that due to the difficulty of separation, the isomers couldn’t be isolated.

### 3.2. The Stability of Pro-7 Is pH Dependent

An in-vitro time-dependent (1–96 h) stability study of Pro-7 was conducted to inform the formulation of the prodrug. The study was conducted at three different pHs: 2.0, 7.4 and 8.0, which mimic stomach, blood and intestinal pH conditions, respectively. The presence of TD-7 or the rate of appearance of TD-7 upon dissociation of Pro-7 was monitored at 349 nm spectrophotometrically. The absorbance versus time was plotted at each buffer system as shown in Figure 3A–C.

Expectedly, the negative control with DMSO showed no presence of TD-7 or Pro-7. The experiment with TD-7 showed stability at all three pHs for the 96 h experiment (Appendix A and Figure 3A,B). The experiment with Pro-7 showed slow degradation of the compound at pHs 7.4 and 8.0, but very fast degradation at pH 2.0 to release TD-7 (Appendix A and Figure 3C). More specifically, it took ~6 h for TD-7 to be detected in a significant amount at pHs 7.4 and 8.0, and only fully dissociated into TD-7 in approximately four days (Figure 3A,B). The formation of TD-7 from Pro-7 at pH 2.0 on the other hand was almost completed within 1 h (Figure 3C). These observations suggest that Pro-7 may degrade quickly into the parent TD-7 in the stomach if given orally, while it may be stable in the other GI parts.

### 3.3. Pro-7 Modifies Hb to Increase the Protein Affinity for Oxygen and Prevents Hypoxia Induced RBC Sickling in a Sustained Manner

Human normal whole blood was used to assess the degree of Hb modification and Hb oxygen affinity (P_50_ shift), while sickle blood was used to investigate RBC sickling inhibition [11,27,36]. Pro-7 and the positive control TD-7 were incubated with normal whole blood suspensions. Aliquot blood samples were drawn and used for assessing Hb adduct formation using HPLC, and Hb oxygen affinity using blood gas analyzer in a time-dependent manner (1–24 h). For the sickling inhibition study, Pro-7 and the positive control TD-7 were incubated with SS blood suspensions under hypoxic conditions (2.5% O_2_/97.5% N_2_), and aliquot blood samples were drawn at three time points (1, 3 and 6 h) and used for assessing RBC morphology. The results summarized in Table 1 and Figure 4A–C show time-dependent effects for all three pharmacodynamic activities. For the Hb modification, the effect of TD-7 peaked at 12 h and then began to gradually decrease, while for the Hb oxygen affinity and RBC sickling inhibition studies, TD-7 peaked quickly at the first hour, and then gradually began to decrease (Figure 4). In contrast, Pro-7 showed a slow but gradual increase in modifying Hb, increasing Hb oxygen affinity and reducing RBC sickling (Figure 4). Even at the end of the 24-h experiment, Pro-7 activity was still increasing. It is clear that in-vitro (using whole blood), the two compounds have different PK/PD profiles, and most importantly Pro-7 showed potency without evidence of metabolic degradation (Figure 4).

### 3.4. Development of Enteric Coated Beads

Solubility of Pro-7 was tested in different solvents to select the most suitable solvent(s) for developing the enteric coated beads. Pro-7 showed an aqueous solubility of 0.165 mg/mL, and solubility of 11.64, 54.89, 66.43, 85.98, and 88.34 mg/mL in acetone, dichloromethane, chloroform, ethanol and methanol, respectively. A mixture of acetone-ethanol (2:1) was used to dissolve Pro-7 and Eudragit. The yield (%) of the enteric coated beads was found to be 89.87%, while the Pro-7 content was 11.03% *w*/*w*. The beads were more stable at pH 1.2 as less than 25% of Pro-7 was released after 2 h, compared to 67% at pH 7.4 within 30 min. Thus, this study informed the selection of acetone-ethanol in a ratio of 2:1 as the most suitable solvent for developing the enteric coating beads of Pro-7.

### 3.5. Development of Intravenous (IV) Formulation

Pro-7 demonstrated a solubility of 68.94 mg/mL when tested within the prepared SNEDDS formulation. To create an IV formulation suitable for animal administration, 1 mL of the Pro-7-loaded SNEDDS formulation was combined with 9 mL of sterile distilled water, and mixed thoroughly. This resulted in an IV formulation containing 6.89 mg Pro-7 per mL water.

Our primary goal was to formulate an IV solution using the SNEDDS approach, aiming for the smallest possible globule size. The SNEDDS formulation exhibited a particle size of 339.83 nm and a polydispersity index (PDI) of 0.611. According to the United States Pharmacopeia (USP) <729> guidelines, it is recommended that the mean globule size of an injectable emulsion should be less than 500 nm [41]. Larger emulsion droplets, with a diameter greater than 5 μm may pose a risk of becoming trapped in narrow blood capillaries, particularly within organs, such as the brain or lungs, potentially leading to embolic complications [42].

### 3.6. Pro-7 Is Capable of Releasing TD-7 in Blood

The PK profile of Pro-7 was assessed in-vivo in male Sprague-Dawley rats following both oral (50 mg/kg) and IV (25 mg/kg) administration of Pro-7. Based on the in-vitro stability study that showed Pro-7 to be unstable at acidic pH, Pro-7 was formulated in enteric-coated formulation for oral administration to help protect the compound from degrading in the highly acidic stomach. Aliquots of blood samples were collected at different time intervals after administration of the Pro-7 to determine the average blood concentrations and the PK parameters (Figure 5). The following parameters were calculated: terminal phase half-life (t½), maximum plasma concentration (C_max_), area under the curve from zero to the last time point (AUC_0–t_), and area under the curve from zero to infinity (AUC_0–α_) (Table 2).

We observed the presence of TD-7 in the bloodstream within the first hour of the experiment following the administration of Pro-7 by oral and IV routes. A C_max_ of 469 ± 322 ng/mL was observed after 0.5 h (T_max_) with the oral Pro-7 treatment group compared to 1426 ± 926 ng/mL for the IV animal group. The hydrolyzed TD-7 was detected in the blood samples of both groups (oral and IV) at the first sampling point, confirming rapid hydrolysis in the blood. In comparison to MX-1520 (thiazolidine prodrug of vanillin), which exhibited a relative bioavailability of 30-fold higher than vanillin, and a corresponding C_max_ 4-fold greater than vanillin (9000 ng/mL and 2000 ng/mL respectively) when administered orally to Sprague-Dawley rats [34], the C_max_ of TD-7 upon administration of Pro-7 (PO and IV) was significantly low to be considered therapeutically relevant. This observation could partly result from the possibility that the enteric coating was not effective in preventing degradation of the thiazolidine moiety into TD-7, leading to its rapid metabolism and subsequent undetectability. However, the most likely explanation points to incomplete absorption due to poor drug solubility and dissolution within the gastrointestinal tract, consistent with the lower AUC values observed in the oral formulation as compared to the IV formulation, with only a fraction of Pro-7 being absorbed (18.4%). Based on these observations, it appears that the enteric coated technique even though may have successfully protected the TD-7 from degradation in the stomach acidic condition, it requires further investigation to enhance the Pro-7 absorption in the gastrointestinal.

## 4. Conclusions

Although TD-7 fits most of the criteria of a potential drug candidate in-vitro, it exhibits short duration of pharmacologic action due to poor oral bioavailability; an undesirable drug property for treating a chronic disease like SCD. We derivatized TD-7 into a thiazolidine prodrug, which is expected to protect TD-7 against oxidative metabolism to improve its PK properties. The prodrug Pro-7 although showed promising in-vitro PD/PK activities, it did not translate into encouraging in-vivo PK outcome. Further investigation to modify or optimize the formulation is warranted to enhance Pro-7 absorption in the gastrointestinal tract, which may improve bioavailability.

## Figures and Tables

**Figure 1 pharmaceutics-15-02547-f001:**
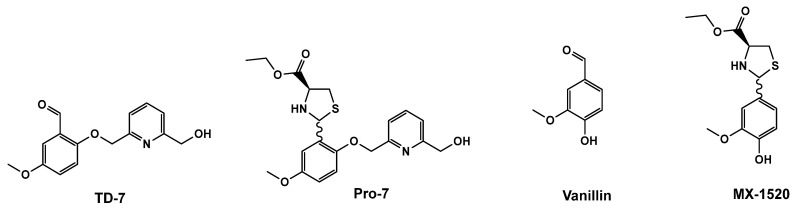
Structures of TD-7 and vanillin and their respective thiazolidine prodrugs.

**Figure 2 pharmaceutics-15-02547-f002:**
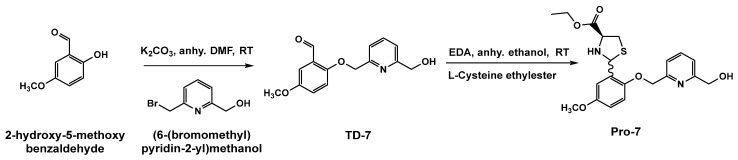
Synthesis of Pro-7.

**Figure 3 pharmaceutics-15-02547-f003:**
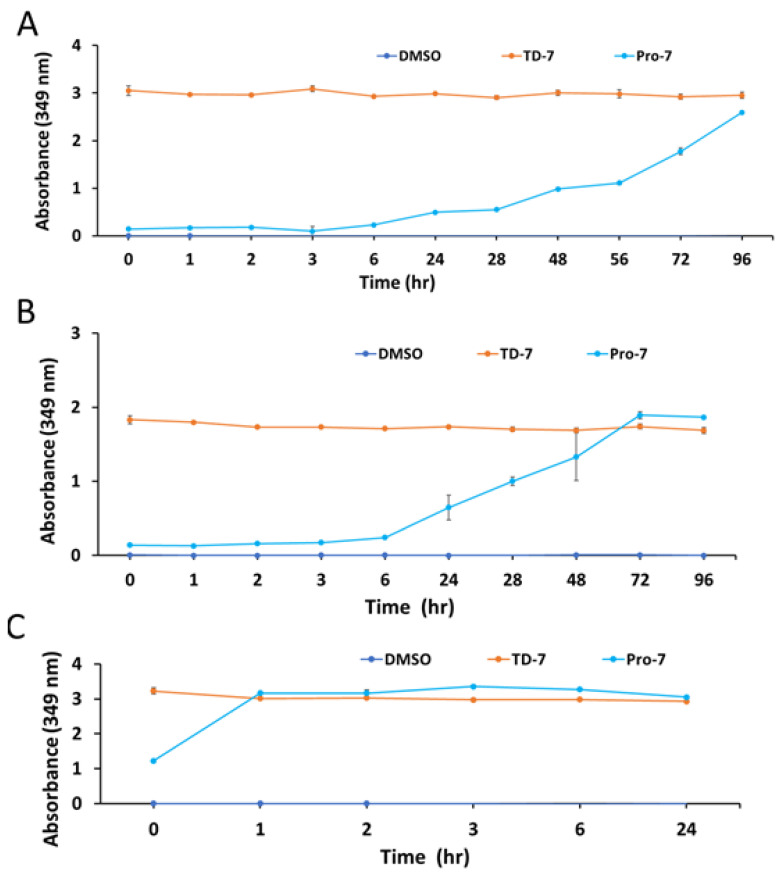
Time-dependent stability studies of Pro-7 and TD-7. (**A**). Stability study at pH 7.0. (**B**). Stability study at pH 8.0. (**C**). Stability study at pH 2.0. All compounds were solubilized in DMSO. The final test compound concentration is 1 mM, and the control experiment without a test compound also included DMSO. The final concentration of DMSO was <2% in all samples, including the control. The results are the mean values of ±SD for biological duplicate measurements.

**Figure 4 pharmaceutics-15-02547-f004:**
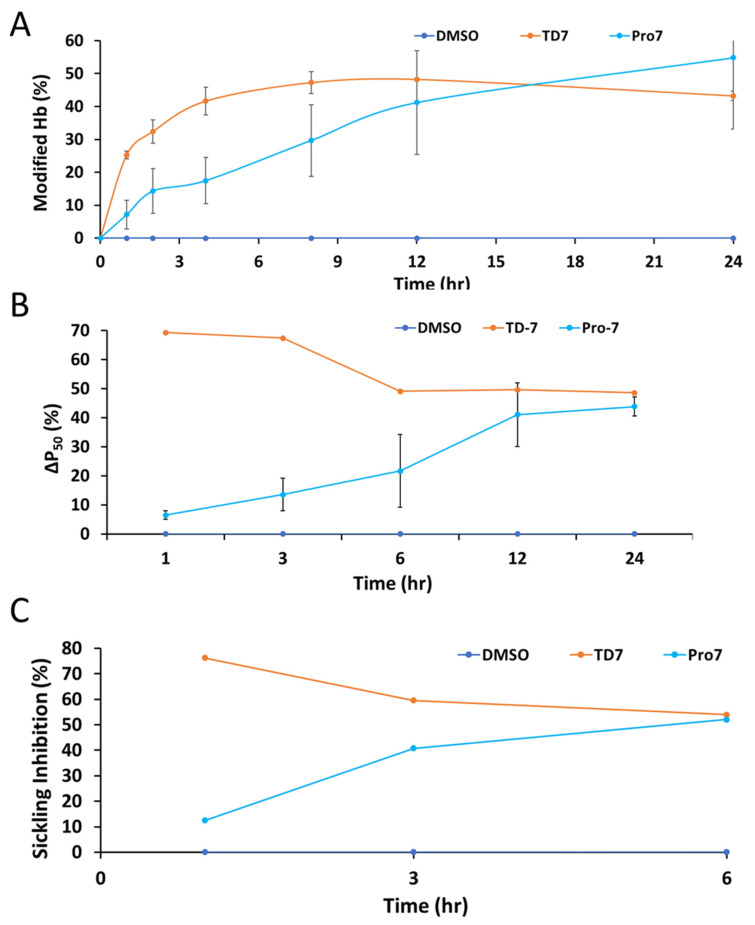
Time-dependent biological/functional effects of TD-7 and Pro-7. (**A**). Hemoglobin modification. (**B**). P_50_ shift. (**C**). RBC sickling inhibition under hypoxia condition. All compounds were solubilized in DMSO. The final test compound concentration is 2 mM. and the control experiment without a test compound also included DMSO. The results are the mean values ± SD for 2 biological replicate experiments. The final concentration of DMSO was <2% in all samples, including in control samples.

**Figure 5 pharmaceutics-15-02547-f005:**
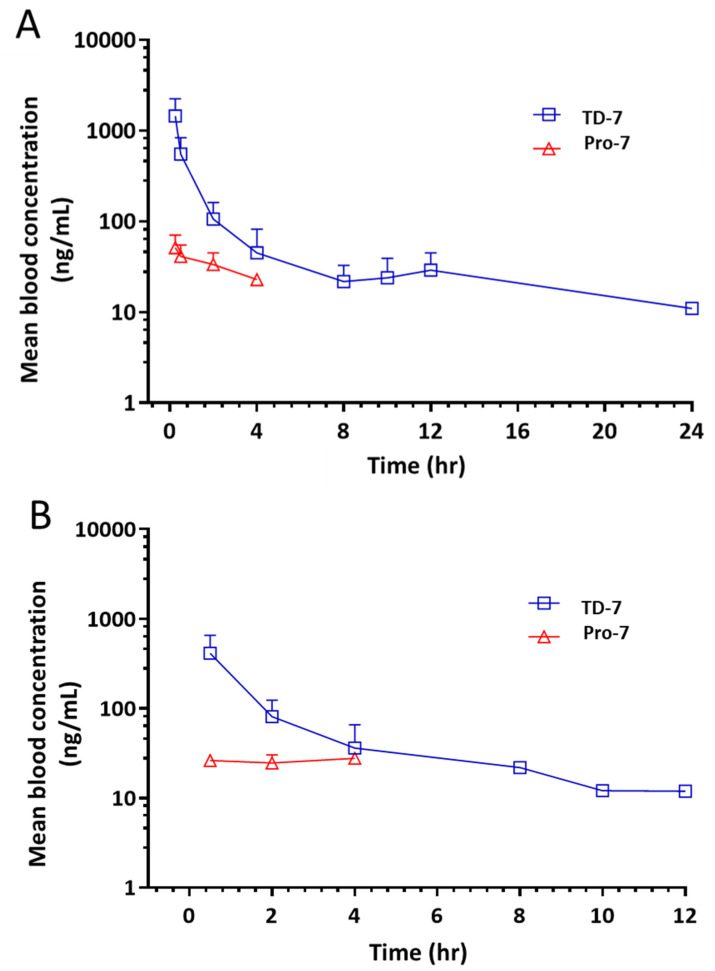
Mean blood concentration versus time profile (*n* = 5, mean ± SD) of TD-7 and Pro-7. (**A**) IV dosing of Pro-7 at 25 mg/kg. (**B**) PO administration of Pro-7 at 50 mg/kg.

**Table 1 pharmaceutics-15-02547-t001:** Hemoglobin modification, change in hemoglobin oxygen affinity, and RBC sickling inhibition studies of TD-7 and Pro-7 ^a^.

	%Hb Modification (Hb Adduct) ^b^			%Hb O_2_ Affinity (P_50_ Shift) ^c^			%Sickling Inhibition ^d^		
Time (h)	TD-7	Pro-7	DMSO	TD-7	Pro-7	DMSO	TD-7	Pro-7	DMSO
0	0.00 ± 0.00	0.00 ± 0.00	0.00	ND	ND	ND	ND	ND	ND
1	25.20 ± 1.20	7.10 ± 4.40	0.00	69.26	6.48 ± 1.54	0.00	76.30	12.53	0.00
2	23.40 ± 3.50	14.30 ± 6.80	0.00	ND	ND	ND	ND	ND	ND
3	ND	ND	ND	67.42	13.57 ± 5.53	0.00	59.62	40.76	0.00
4	41.70 ± 4.30	17.50 ± 7.10	0.00	ND	ND	ND	ND	ND	ND
6	ND	ND	ND	49.06	21.64 ± 10.93	0.00	54.02	52.04	0.00
8	47.30 ± 4.70	29.70 ± 10.9	0.00	ND	ND	ND	ND	ND	ND
12	48.30 ± 0.50	41.20 ± 15.8	0.00	49.58	41.04 ± 10.93	0.00	ND	ND	ND
24	43.30 ± 1.90	54.80 ± 21.8	0.00	48.56	43.87 ± 3.22	0.00	ND	ND	ND

^a^ The Hb modification and Hb oxygen affinity studies were conducted with human normal whole blood (30% hematocrit), while the sickling inhibition study conducted with SS cells suspensions (20% hematocrit) incubated with 2 mM of TD-7 or Pro-7. The results are the mean values ±SD for 2 biological replicate experiments. The final concentration of DMSO was <2% in all samples, including in control samples. ^b^ Hb adduct or modification values obtained from HPLC elution patterns of individual hemolysates after incubation of compounds with normal blood. ^c^ P_50_ is the oxygen pressure at which Hb in normal whole blood are 50% saturated with oxygen. $\Delta P_{50}\left(\%\right)=\frac{P_{50}\;\mathrm{of}\;\mathrm{lysates}\;\mathrm{from}\;\mathrm{untreated}\;\mathrm{cells}\;–{\;P}_{50}\;\mathrm{of}\;\mathrm{lysates}\;\mathrm{from}\;\mathrm{treated}\;\mathrm{cells}}{P_{50}\;\mathrm{of}\;\mathrm{lysates}\;\mathrm{from}\;\mathrm{untreated}\;\mathrm{cells}}\;\times \;100$. ^d^ RBC sickling inhibition studies with SS cells were conducted under hypoxia (4% oxygen/96% nitrogen).

**Table 2 pharmaceutics-15-02547-t002:** Pharmacokinetic parameters of TD-7 and Pro-7 following oral (50 mg/kg) and intravenous administration (25 mg/kg) of Pro-7 PO to male Sprague-Dawley rats (*n* = 5).

Pharmacokinetic Parameter	Intravenous Administration	Oral Administration
t½ (h)	4.98 ± 1.89	4.73 ± 1.22
C_max_ (ng/mL)	1426 ± 926	469 ± 322
T_max_ (h)	-	0.5 ± 0.0
AUC_0–t_ (ng∙h/mL)	1830 ± 737	672 ± 255
AUC_0–∞_ (ng∙h/mL)	1956 ± 792	730 ± 225
Cl (mL/min/kg)	238 ± 85.9	200.89 ± 34.3
V_d_ (L/kg)	98.9 ± 45.3	83.7 ± 22.1
Fraction of drug absorbed (% F)	-	18.4

## Data Availability

Not applicable.

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
