# Peer review of "Design, Synthesis, and Antisickling Investigation of a Thiazolidine Prodrug of TD-7 That Prolongs the Duration of Action of Antisickling Aromatic Aldehyde"

_pharmaceutics, 2023, doi:10.3390/pharmaceutics15112547_

Round 1
Reviewer 1 Report
Comments and Suggestions for Authors
Rana T an this colleagues from Saudi Arabia report on a new drug TD-7 as a potential antisickling fro sickle cell anemia. This is an experimental study very well designed and performed.
The results are relevant and deserve to be published
This review however recommend that methods should be revised by a biochemistry consultant qualified in haemoglobin disorders
Minor corrections:
INTRODUCTION
- after Ref 12, add these ref:
Insight into the complex pathophysiology of sickle cell anaemia and possible treatment.
Piccin A, et al. Eur J Haematol. 2019.
Sickle cell disease and dental treatment.
Piccin A, et al. J Ir Dent Assoc. 2008
The 'scintilla' starting vaso-occlusion in sickle cell disease.
Piccin A, et al. Br J Haematol. 2023
METHOD:
- delete line 91: "as a kind of gift"
- METHODS
Author Response
Reviewer 1
Rana T an this colleagues from Saudi Arabia report on a new drug TD-7 as a potential antisickling from sickle cell anemia. This is an experimental study very well designed and performed.
The results are relevant and deserve to be published
This review however recommend that methods should be revised by a biochemistry consultant qualified in haemoglobin disorders
Minor corrections:
INTRODUCTION
- After Ref 12, add these ref:
Insight into the complex pathophysiology of sickle cell anaemia and possible treatment.
Piccin A, et al. Eur J Haematol. 2019.
Sickle cell disease and dental treatment.
Piccin A, et al. J Ir Dent Assoc. 2008
The 'scintilla' starting vaso-occlusion in sickle cell disease.
Piccin A, et al. Br J Haematol. 2023
Reply
We have cited the first and third references in the introduction
METHOD:
- delete line 91: "as a kind of gift"
Reply
The phrase has been deleted.
Reviewer 2 Report
Comments and Suggestions for Authors
The publication submitted for evaluation concerns the synthesis and research on thiazolidine prodrug of TD-7. The manuscript is interesting although the research results are not satisfactory.
Despite the lack of satisfactory results, the work constitutes a contribution to the optimization of the described structure towards its application in the treatment of sickle cell anemia.
In my opinion, such results do not disqualify for publication. However, the synthesis and purity of the tested substance are of great concern. Excesses of some reagents are used in the synthesis (why in such proportions?), after completing the synthesis, the authors evaporate the solvent and obtain a pure substance. Without cleansing? - This is clear from the description of the synthesis.
The authors performed in vivo tests on the substance obtained in this way?
Moreover, a correctly characterized new organic substance should have 1H NMR spectrum signals assigned to the appropriate protons. You should number the atoms in the molecule and then describe them from the smallest to the largest shift.
The melting point is given as a range - start of melting - end of melting (the entire solid is a liquid). This range allows, in many cases, to initially assess the purity of the substance. When describing the MS analysis, the mass of the substance and its summary formula should be provided.
Degradation on the C-8 column does not explain the indeterminacy of the purity of the substance. Purity can also be determined by elemental analysis or HRMS analysis. MNR analyzes show that the substance is contaminated.
Author Response
Reviewer 2
The publication submitted for evaluation concerns the synthesis and research on thiazolidine prodrug of TD-7. The manuscript is interesting although the research results are not satisfactory. Despite the lack of satisfactory results, the work constitutes a contribution to the optimization of the described structure towards its application in the treatment of sickle cell anemia. In my opinion, such results do not disqualify for publication. However, the synthesis and purity of the tested substance are of great concern. Excesses of some reagents are used in the synthesis (why in such proportions?), after completing the synthesis, the authors evaporate the solvent and obtain a pure substance. Without cleansing? - This is clear from the description of the synthesis. The authors performed in vivo tests on the substance obtained in this way?
- Excesses of some reagents are used in the synthesis (why in such proportions?)
Reply
The thiazolidine prodrug of TD-7, Pro-7 was synthesized following the published procedure for synthesizing the thiazolidine prodrug of vanillin, MX-150. We have cited this work in the synthesis section (doi:10.1111/j.1365-2141.2004.04892.x).
- After completing the synthesis, the authors evaporate the solvent and obtain a pure substance. Without cleansing? The authors performed in vivo tests on the substance obtained in this way?
Reply
We have revised the chemistry section to include the following detailed purification step. “The reaction mixture was quenched with water, and the resulting white precipitate was subsequently isolated through filtration, followed by several washing cycles with water. The dried material was crystallized using methanol, and the crystals filtered using vacuum drying with a final product yield of 40.9%.”
- Moreover, a correctly characterized new organic substance should have 1H NMR spectrum signals assigned to the appropriate protons. You should number the atoms in the molecule and then describe them from the smallest to the largest shift.
Reply
We have revised the 1H NMR spectrum, an updated 1H NMR spectrum has been submitted as supplementary data. Also, we have revised the chemistry section (result and discussion) and described the protons from the smallest to the largest shift.
- The melting point is given as a range - start of melting - end of melting (the entire solid is a liquid). This range allows, in many cases, to initially assess the purity of the substance. When describing the MS analysis, the mass of the substance and its summary formula should be provided.
Reply
The correction is made to include the melting point range of 100.8 °C–101.5 °C.
We have updated the synthesis section and provided the mass of the substance and its formula C20H24N2O5S (404.4810).
- Degradation on the C-8 column does not explain the indeterminacy of the purity of the substance. Purity can also be determined by elemental analysis or HRMS analysis. MNR analyzes show that the substance is contaminated.
Reply
As noted in the manuscript, the use of HPLC to determine the purity of Pro-7 did not work because of degradation of the compound on the C-8 column. Detailed explanation is now provided in the manuscript.
Reviewer 3 Report
Comments and Suggestions for Authors
Current report demonstrated the synthesis of a thiazolidine prodrug of TD-7, referred to as Pro-7, along with a comprehensive investigation of Pro-7 functional and biological properties. I like to give the following comments.
1. Statistical analysis was not included in Materials and Methods. Why?
2. Pharmacokinetic data analysis needs the reference(s) to support.
3. In stability study, preparation of Pro-7 or TD-7 in solution must describe in detail.
4. In Table 2 and many figures, sample size was not indicated.
5. Chemical similarity between Pro-7 and MX-1520 was not discussed. Why?
6. In conclusion, tweaking of the formulation has been suggested as the way to solve the problem of Pro-7. However, how to do it was not described in clear.
Comments on the Quality of English LanguageIt seems better to check through the professional editing.
Author Response
Reviewer 3
Current report demonstrated the synthesis of a thiazolidine prodrug of TD-7, referred to as Pro-7, along with a comprehensive investigation of Pro-7 functional and biological properties. I like to give the following comments.
- Statistical analysis was not included in Materials and Methods. Why?
Reply
Statistical analysis section has now been included.
- Pharmacokinetic data analysis needs the reference(s) to support.
Reply
Reference has been added to the revised manuscript.
- In stability study, preparation of Pro-7 or TD-7 in solution must describe in detail.
Reply
Details about the preparation of Pro-7 and TD-7 have been added to the revised manuscript.
- In Table 2 and many figures, sample size was not indicated.
Reply
Sample size has been added to all tables and figures.
- Chemical similarity between Pro-7 and MX-1520 was not discussed. Why?
Reply
We have revised the introduction with the following cited statement.
“TD-7 is a second-generation aromatic aldehyde, obtained by incorporating an alcohol substituted methoxypyridine to the benzaldehyde ring of a vanillin analog (Figure 1), resulting in TD-7 exhibiting several fold sickling inhibition potency over vanillin.”
In this study, we have used the thiazolidine prodrug approach, where the aldehyde group of TD-7 was protected via a coupling reaction with L-cysteine ethyl ester to form a thiazolidine complex Pro-7, enhancing the metabolic stability of TD-7.
- In conclusion, tweaking of the formulation has been suggested as the way to solve the problem of Pro-7. However, how to do it was not described in clear.
Reply
The conclusion section has been appropriately modified.
Comments on the Quality of English Language: It seems better to check through the professional editing.
Reply
We have made several grammatical corrections to improve the presentation of the manuscript.
Reviewer 4 Report
Comments and Suggestions for Authors
In the manuscript by Rana T. Alhashimi et al. entitled “Design, Synthesis, and Antisickling Investigation of a Thiazolidine Prodrug of TD-7 that Prolongs the Duration of Action of Antisickling Aromatic Aldehyde”; the authors explored TD-7, an allosteric effector of hemoglobin, as a treatment for sickle cell disease. They synthesized a thiazolidine prodrug, Pro-7, and investigated its properties. In vitro studies showed Pro-7's gradual but increasing effectiveness in modifying hemoglobin and affecting oxygen affinity. In vivo studies with rats demonstrated Pro-7's potential to release TD-7 through hydrolysis, though therapeutic blood levels were not reached. This research suggests that the addition of the L-cysteine ethyl ester group to TD-7 may enhance the stability of aromatic aldehydes, offering promise for more effective sickle cell disease treatment.
However, it's important to note that further investigation is needed, and the manuscript requires revisions.
General comments:
The use of 9 keywords seems excessive; reducing them to 5 or 6 would enhance the clarity and conciseness of the paper.
Moreover, the references should be positioned before commas (,) or at the end of sentences (before the points), in line with the standard referencing style.
Additionally, it's essential to ensure that the list of abbreviations is placed at the end of the manuscript, following the Pharmaceutics template.
The modification of 'Sheme 1' to 'Figure 2' and the consequent renumbering of other figures, as well as cross-referencing adjustments in the text, should be made to maintain consistency and improve the manuscript's flow.
Materials and methods section:
Consider incorporating a detailed statistical analysis to strengthen the overall quality and credibility of your research. Statistical analysis is a crucial component of scientific research, and it's essential for ensuring the rigor and reliability of the study's findings. Adding a statistical section would not only enhance the transparency of your methodology but also allow readers to better understand the data analysis processes and the significance of the results.
The authors should be mindful of the cytotoxic effects of DMSO on red blood cells in their in vitro studies. To mitigate potential harm, considering an alternative solvent that is less harmful to cells or justifying the use of DMSO while outlining precautions taken is essential. Ensuring transparency in the manuscript about the solvent choice, limitations, and safety measures will enhance the research's scientific rigor and ethical standards.
It is crucial to include a section detailing the animal housing conditions in your manuscript. By providing this information, you not only enhance the transparency of your research but also ensure that your findings can be properly contextualized and compared with other studies.
In the in vivo study (section 2.7.), the authors should consider including three additional groups to further validate and enhance the robustness of their findings:
Normal Control Group: This group should consist of subjects that did not receive any administration, serving as a baseline reference.
Control Group for IV Administration: A control group for IV administration should be included, and it is essential that the authors clarify the vehicle used for this group. Administering only the vehicle via the same route is crucial for distinguishing the effects of the substance under investigation from the potential effects of the delivery method.
Control Group for Oral Administration: Similarly, a control group for oral administration should be established.
Discussion and Results section:
In Figure 3B, it is crucial to include the time (in hours) on the x-axis for clarity. Additionally, please ensure that the presence or absence of DMSO is clearly represented in all relevant figures to provide a complete understanding of the experimental conditions throughout the manuscript.
Comments on the Quality of English Language
I noticed a few typos and grammatical errors in the text. It requires minor editing for the English language.
Author Response
Reviewer 4
In the manuscript by Rana T. Alhashimi et al. entitled “Design, Synthesis, and Antisickling Investigation of a Thiazolidine Prodrug of TD-7 that Prolongs the Duration of Action of Antisickling Aromatic Aldehyde”; the authors explored TD-7, an allosteric effector of hemoglobin, as a treatment for sickle cell disease. They synthesized a thiazolidine prodrug, Pro-7, and investigated its properties. In vitro studies showed Pro-7's gradual but increasing effectiveness in modifying hemoglobin and affecting oxygen affinity. In vivo studies with rats demonstrated Pro-7's potential to release TD-7 through hydrolysis, though therapeutic blood levels were not reached. This research suggests that the addition of the L-cysteine ethyl ester group to TD-7 may enhance the stability of aromatic aldehydes, offering promise for more effective sickle cell disease treatment.
However, it's important to note that further investigation is needed, and the manuscript requires revisions.
General comments:
- The use of 9 keywords seems excessive; reducing them to 5 or 6 would enhance the clarity and conciseness of the paper.
Reply
The keywords have been reduced to 6 in the modified manuscript.
- Moreover, the references should be positioned before commas (,) or at the end of sentences (before the points), in line with the standard referencing style.
Reply
The references have been fixed.
- Additionally, it's essential to ensure that the list of abbreviations is placed at the end of the manuscript, following the Pharmaceutics template.
Reply
The list of abbreviations has been moved to the end of the manuscript.
- The modification of 'Scheme 1' to 'Figure 2' and the consequent renumbering of other figures, as well as cross-referencing adjustments in the text, should be made to maintain consistency and improve the manuscript's flow.
Reply
The necessary changes have been made.
Materials and methods section:
- Consider incorporating a detailed statistical analysis to strengthen the overall quality and credibility of your research. Statistical analysis is a crucial component of scientific research, and it's essential for ensuring the rigor and reliability of the study's findings. Adding a statistical section would not only enhance the transparency of your methodology but also allow readers to better understand the data analysis processes and the significance of the results.
Reply
Statistical analysis part has been added to the revised manuscript.
- The authors should be mindful of the cytotoxic effects of DMSO on red blood cells in their in vitro studies. To mitigate potential harm, considering an alternative solvent that is less harmful to cells or justifying the use of DMSO while outlining precautions taken is essential. Ensuring transparency in the manuscript about the solvent choice, limitations, and safety measures will enhance the research's scientific rigor and ethical standards.
Reply
Thank you for pointing out the potential adverse effect of DMSO may have on red blood cells. We and others have used DMSO for in vitro studies with red blood cells, and at concentration of less than 2%, no significant cytotoxic effect is observed. In this study the final concentration of DMSO was <2%.
It is crucial to include a section detailing the animal housing conditions in your manuscript. By providing this information, you not only enhance the transparency of your research but also ensure that your findings can be properly contextualized and compared with other studies.
Reply
Animal housing section has been added to the revised manuscript.
- In the in vivo study (section 2.7.), the authors should consider including three additional groups to further validate and enhance the robustness of their findings:
Normal Control Group: This group should consist of subjects that did not receive any administration, serving as a baseline reference.
Control Group for IV Administration: A control group for IV administration should be included, and it is essential that the authors clarify the vehicle used for this group. Administering only the vehicle via the same route is crucial for distinguishing the effects of the substance under investigation from the potential effects of the delivery method.
Control Group for Oral Administration: Similarly, a control group for oral administration should be established.
Reply
We completely agree with the reviewer comment raised regarding the control groups, and we would like to extend our sincere apologies for omitting this crucial information in our original submission. The requested details have been added to the revised manuscript.
Discussion and Results section:
- In Figure 3B, it is crucial to include the time (in hours) on the x-axis for clarity. Additionally, please ensure that the presence or absence of DMSO is clearly represented in all relevant figures to provide a complete understanding of the experimental conditions throughout the manuscript.
Reply
We have revised the figures and ensured the time is indicated by hours, also we made sure of the illustration of DMSO presence or absence in the figures.
Comments on the Quality of English Language: I noticed a few typos and grammatical errors in the text. It requires minor editing for the English language.
Reply
We have made appropriates grammatical corrections throughout the manuscript.
Round 2
Reviewer 2 Report
Comments and Suggestions for Authors
Thank you for your insightful response to the review. In my opinion, the manuscript can be published.
Reviewer 4 Report
Comments and Suggestions for Authors
I have reviewed the revised manuscript and confirm that the authors have addressed all raised issues. The manuscript has been significantly improved and is now suitable for publication in its current form.